**∂ | Open Peer Review** | Bacteriology | Research Article

# Expansion of the Group B *Streptococcus* serotype repertoire via gene acquisition from other streptococcal species

Molly E. Sharp,[1] Julia Sproch,[1] Sydney Haldeman,[2] Hervé Tettelin,[3] Adam J. Ratner[1,2]

**ABSTRACT** Group B *Streptococcus* (GBS) is a major cause of invasive infection in infants. The leading GBS vaccine candidate is a capsular polysaccharide-protein conjugate vaccine based on the six most common disease-causing serotypes (Ia, Ib, II–V). Four more recently discovered, less common serotypes (VI–IX) also circulate in the human population. Serotype VIII was initially described in the 1980s in Japan, where it made up a significant proportion of colonizing isolates in pregnant patients. Serotype VIII continues to be an emerging cause of colonization and disease globally. In addition to the 10 known GBS serotypes, intra- and interspecies horizontal gene transfer (HGT) could create GBS strains with novel capsule structures, potentially leading to vaccine escape. Previous work speculated that serotype VIII might be the result of interspecies HGT of a portion of the *cps* locus. We investigated the function and potential sources of *cpsR*, encoding a rhamnosyltransferase, in serotype VIII GBS. In a broad-based search for CpsR orthologs, proteins from streptococcal species that live in niches overlapping with GBS (including *S. suis* and *S. gallolyticus*) were closely related to CpsR. An unmarked, in-frame GBS Δ*cpsR* mutant was no longer recognized by serotype VIII-specific antibodies. Reactivity was restored by expressing wild-type *cpsR* or orthologs from *S. suis* and *S. gallolyticus*. In a murine model of vaginal co-colonization, the Δ*cpsR* mutant was outcompeted by wild-type serotype VIII, suggesting functional serotype VIII capsule provides a competitive advantage *in vivo*. Our findings are consistent with interspecies HGT as a mechanism underlying emergence of serotype VIII GBS.

**IMPORTANCE** Capsular polysaccharide (CPS) is a key virulence factor that aids group B *Streptococcus* (GBS) in colonization and pathogenicity in humans. The major human vaccine candidate against GBS is a CPS-based vaccine including six serotypes. In addition to the four non-vaccine serotypes, intra- and interspecies horizontal gene transfer (HGT) could create GBS strains with novel capsule structures, potentially leading to vaccine escape. Here, we describe a key gene, *cpsR*, in the production of GBS serotype VIII and the complementation of its function by orthologs from other streptococci. Our work demonstrates GBS's ability to utilize genes from other streptococci to produce functional capsular polysaccharide. HGT could generate further capsule diversity beyond the 10 current serotypes, potentially increasing the number of strains capable of vaccine escape. Surveillance for such events is warranted.

**KEYWORDS** *Streptococcus agalactiae* (Group B *Streptococcus*), capsule serotype, genetic exchange

G roup B *Streptococcus* (GBS) is a leading cause of neonatal disease worldwide (1, 2). GBS colonizes the gastrointestinal tract and can migrate to and asymptomatically colonize the vaginal tract (3, 4). Colonization of the vaginal tract during pregnancy can lead to neonatal invasive infection resulting in meningitis and sepsis (5, 6). GBS is also emerging as a cause of invasive disease in non-pregnant adults (7–9).

**Peer Reviewer** Ryan S. Doster, University of Louisville, Louisville, Kentucky, USA

Address correspondence to Adam J. Ratner, Adam.Ratner@nyulangone.org.

The authors declare no conflict of interest.

See the funding table on p. 13.

Nearly all strains of GBS produce a polysaccharide capsule that coats the bacteria and is covalently attached to the peptidoglycan (10). Sialylation of the capsular polysaccharide (CPS) in GBS enhances virulence and helps in evading the immune system (11–13). Encapsulation aids the bacteria in vaginal and gastrointestinal colonization (14, 15), as well as ascending infection (16) and survival in human blood (17).

GBS has 10 known serotypes (Ia, Ib, II, III, IV, V, VI, VII, VIII, and IX), defined by specific CPS structures. The prevalence of each serotype varies by geographical region and disease type. Serotypes Ia, Ib, II, III, IV, and V comprise 98% of invasive disease isolates in the United States (18), 95% of disease isolates in pregnant patients worldwide (19), and over 96% of invasive disease in infants in Europe (20). Because of this, the capsular polysaccharide-protein conjugate vaccine currently in phase II trials covers serotypes Ia through V (21). Worldwide colonization rates are similar, with serotypes Ia–V accounting for 98% of isolates obtained during pregnancy, but regional studies show that Eastern and South-Eastern Asia have up to 20% colonization rates of non-vaccine-type serotypes VI–IX (22).

Serotype VIII GBS was first characterized in 1989 and retrospectively identified in samples in Japan dating back to 1979 (23). Throughout the 1990s, colonization rates of serotype VIII in pregnant patients were up to 35.6% in Japan (24, 25), while remaining quite rare in colonization and disease in other countries (26–28). These rates began to decrease in Japan in the 2000s (25, 29), but serotype VIII has emerged in other areas of the world and remains common in Eastern and South-Eastern Asia (22). More recently, a Korean study found 20% colonization rates of serotype VIII in pregnant patients (30). Serotype VIII GBS strains can cause invasive disease in neonates and adults (31–35). In 2023, serotype VIII was identified as an emerging cause of invasive disease in Alberta, Canada (31). Serotype VIII is not currently included in the polysaccharide-based vaccine, creating a potential threat for serotype replacement post-vaccination.

Serotype VIII is the only GBS serotype to contain rhamnose and encode a rhamnosyltransferase, CpsR (36). Cieslewicz et al. found that, in addition to *cpsR,* the serotype VIII *cpsE, cpsH, cpsJ,* and *cpsK* genes were most distantly related to the glycosyltransferases of other GBS serotypes. Because the serotype VIII structure is similar to that of *Streptococcus pneumoniae* serotype 23F, they hypothesized either that these genes were acquired by horizontal gene transfer (HGT) from *S. pneumoniae* or that both received DNA from some other species (37). Here, we demonstrate CpsR is a key glycosyltransferase in constructing the specific serotype VIII CPS structure. We show that serotype VIII capsule provides a competitive advantage in an *in vivo* model of murine vaginal colonization and that the function of CpsR can be replaced by rhamnosyltransferases from other streptococcal species. These findings further support the theory that GBS acquired the type VIII capsule through HGT, showcasing a mechanism for generation of GBS CPS diversity and a potential threat to conjugate vaccine efficacy.

## RESULTS

### *cpsR* is a serotype VIII-specific GBS *cps* gene

Ten serotypes of GBS have been identified. Among these, only serotype VIII produces a CPS structure that includes rhamnose as one of its monosaccharides (Fig. 1A). The gene *cpsR*, which encodes a rhamnosyltransferase, is present only in the serotype VIII *cps* locus (Fig. 1B). To understand the degree of difference between the serotype VIII *cps* locus and the rest of the GBS serotypes, we aligned the nucleotide sequences of each gene in the serotype VIII *cps* locus to its cognate gene in the other nine serotypes using ClustalW. The reference serotype VIII sequence is from the Centers for Disease Control and Prevention's (CDC) Active Bacterial Core surveillance (ABCs) and is referred to in our collection as AR977 (all strains from the CDC are reported in Table S1). Percent identities from these alignments are reported in Fig. 2. As has previously been reported, the 5′ and 3′ ends of the *cps* locus are well conserved across serotypes. *cpsABCD*, *cpsL*, and *neuBCDA* all had nucleotide identities over 92%. Besides *cpsR*, which has no cognate gene in other GBS *cps* loci, the genes in the middle of the locus, *cpsE*, *cpsH*, *cpsJ*, and *cpsK,* all had nucleotide

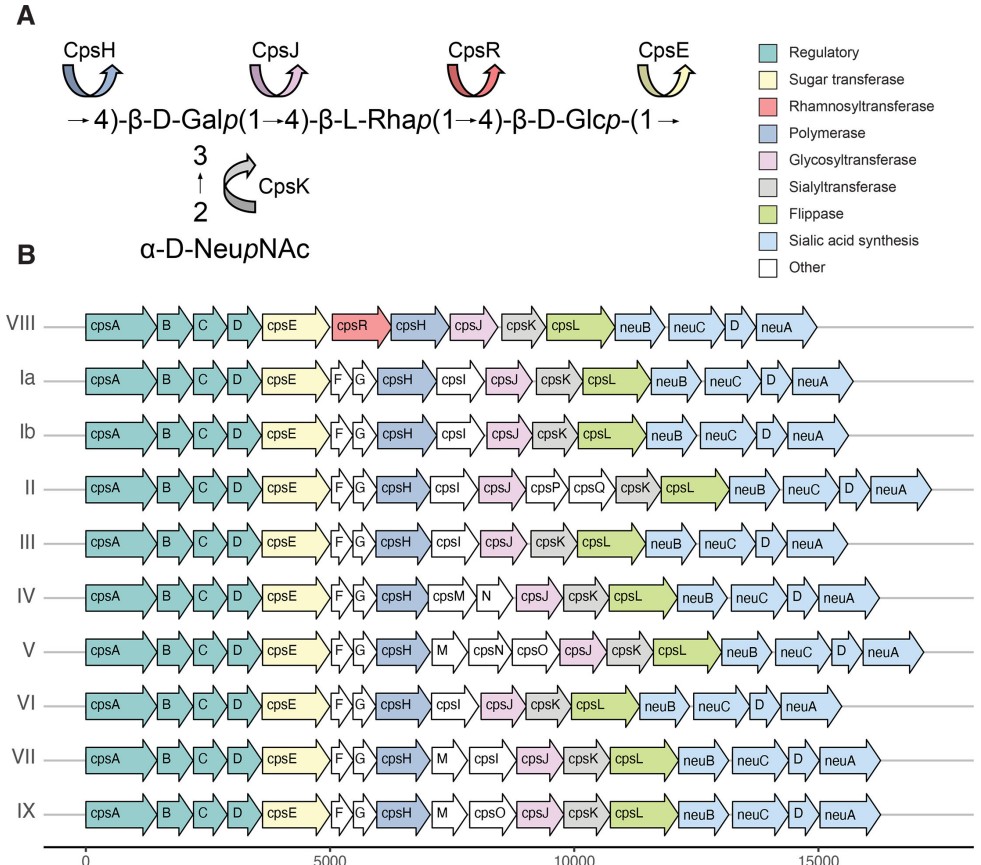

**FIG 1** *cpsR* encodes a rhamnosyltransferase that is specific to GBS serotype VIII. (A) Chemical structure of the repeating unit of the serotype VIII capsular polysaccharide (36). The putative functions of enzymes are represented by arrows with colors corresponding to the gene encoding them in Fig. 1B. Putative function is based on previously reported sequence alignments. (B) Schematic of the 10 GBS *cps* loci based on representative sequences from each serotype (38). The serotype VIII wild-type locus used in this study, AR977, is displayed at the top. Color fill of arrows represents putative conserved function between serotypes as compared to serotype VIII, not amino acid or nucleic acid sequence alignment.

identities under 55%. Similar to Cieslewicz et al., we concluded that the portion of the *cps* locus, *cpsERHJK*, was likely horizontally transferred to GBS, contributing to the evolution of serotype VIII.

## CpsR and surrounding proteins are closely related to other streptococcal species

To investigate potential donors of genetic material to GBS, we used basic local alignment search tool (BLASTp) to search the National Center for Biotechnology Information's (NCBI) non-redundant protein sequences (nr) database for proteins with similar amino acid sequences to GBS serotype VIII. We focused on the proteins encoded in the region of interest of the *cps* locus identified in Fig. 2 (*cpsERHJK*). We individually aligned the proteins CpsE, CpsR, CpsH, CpsJ, and CpsK and reported the bacterial species containing the protein with the most similar amino acid sequence in Table 1 (the top 100 sequence alignments are reported in Table S2). CpsE was most similar to proteins from *Streptococcus suis* and *Streptococcus orisratti*, with amino acid percent identities of 64.2%. CpsR was also closest in sequence to a protein from *S. suis*, with a 78.0% identity. CpsH was most closely aligned to a protein from *Streptococcus gallolyticus,* with a percent identity of 69.4%. CpsJ was most closely aligned to proteins from *Streptococcus equinus* and *S. gallolyticus,* whose sequences shared a percent identity of 64.6%. CpsK shared less than 42% amino acid sequence identity with proteins from any other species.

| | cpsA | cpsB | cpsC | cpsD | cpsE | cpsR | cpsH | cpsJ | cpsK | cpsL | neuB | neuC | neuD | neuA |
|----|------|------|------|------|------|------|------|------|------|------|------|------|------|------|
| Ia | 99.7 | 99.5 | 99.4 | 97.9 | 52.5 | - | 39.2 | 45.0 | 38.3 | 99.1 | 98.9 | 99.4 | 99.4 | 98.9 |
| Ib | 99.6 | 99.3 | 99.1 | 97.9 | 52.8 | - | 38.1 | 38.6 | 39.9 | 98.3 | 98.7 | 99.0 | 99.2 | 99.5 |
| II | 99.8 | 99.5 | 99.6 | 99.0 | 52.9 | - | 36.4 | 48.6 | 33.4 | 98.5 | 99.1 | 99.5 | 99.5 | 99.0 |
| III | 99.2 | 98.8 | 99.4 | 97.1 | 53.1 | - | 36.0 | 45.0 | 38.3 | 99.0 | 99.0 | 99.1 | 99.4 | 99.5 |
| IV | 99.4 | 99.3 | 99.3 | 98.9 | 52.9 | - | 36.5 | 46.3 | 39.2 | 97.6 | 99.6 | 99.6 | 99.8 | 99.6 |
| V | 99.6 | 99.5 | 99.7 | 99.0 | 52.9 | - | 37.1 | 45.2 | 38.0 | 97.6 | 99.2 | 99.5 | 99.5 | 99.1 |
| VI | 99.1 | 99.3 | 99.1 | 97.4 | 53.2 | - | 37.0 | 40.3 | 38.8 | 98.7 | 99.5 | 99.4 | 99.5 | 99.0 |
| VII | 99.8 | 99.5 | 99.4 | 99.1 | 53.0 | - | 37.1 | 45.5 | 37.1 | 92.0 | 97.4 | 99.5 | 99.5 | 99.0 |
| IX | 99.2 | 99.2 | 99.0 | 99.0 | 53.1 | - | 37.1 | 44.7 | 38.1 | 97.0 | 99.1 | 99.6 | 99.4 | 99.4 |

**FIG 2** Nucleotide sequence percent identities between serotype VIII *cps* locus and corresponding genes in the other nine GBS serotypes. Degree of similarity is represented in gray scale. *cpsR* is absent in all serotypes besides VIII.

Based on these results, we focused on *S. suis* and *S. gallolyticus* as potential candidates for a genetic transfer event. *S. suis* and *S. gallolyticus* were selected because they each had the most similar sequences for two out of the four Cps proteins and were present in the top 100 alignments for all four proteins. We therefore aligned the *cps* locus of GBS serotype VIII to the *cps* loci of *S. suis* and *S. gallolyticus* to assess the similarities of the proposed region of transfer. Figure 3A shows the nucleotide identities between genes from wild-type VIII strain AR977 and genes from the *S. suis* and *S. gallolyticus* strains identified as most similar in the Cps BLASTp results. *cpsR* has over 70% nucleotide sequence identity with *S. suis cpsF*, while the downstream genes *cpsH* and *cpsJ* had over 70% sequence identity with *S. gallolyticus*. This supports our theory that an interspecies genetic transfer of this region of the *cps* locus might have contributed to the generation of serotype VIII.

In addition, we created a phylogenetic tree of GBS CpsR orthologs using 820 of the most similar amino acid sequences (Fig. 3B and C). We focus on CpsR for this tree because it is the only protein encoded in the serotype VIII *cps* locus that is not present in other GBS *cps* loci and, therefore, is most likely to have been transferred to GBS from another species. The 100 sequences from our CpsR BLASTp results were aligned using MUSCLE v.3.7 and used to create a hidden Markov model (HMM) that was used to search the NCBI nr database for additional orthologs. Eight hundred twenty sequences with a bit score >700 were chosen and aligned with the wild-type serotype VIII CpsR sequence. This alignment was used to build a maximum-likelihood phylogenetic tree. The tree shows the evolutionary relationship of GBS CpsR to other streptococcal rhamnosyltransferases (Fig. 3B). In agreement with our BLASTp results, the closest relatives of GBS CpsR are rhamnosyltransferases from *S. suis* (Fig. 3C). Additional sequences from *S. suis* and *S. gallolyticus* both appear in the subsetted tree in Fig. 3C along with sequences from other streptococcal species.

## *cpsR* is essential for serotype VIII CPS production

To investigate the importance of *cpsR* for serotype VIII capsule production, we created an in-frame deletion of *cpsR* in a clinical isolate of GBS serotype VIII, AR977. Mutation was achieved via homologous recombination, resulting in deletion mutants or revertant strains genetically identical to wild-type, which were used as controls throughout this work. We also created a ΔcpsE deletion mutant. *cpsE* is required for capsule production in multiple GBS serotypes (14, 16, 39). The ΔcpsE mutant was used as an acapsular control. When we performed a latex agglutination assay using Immulex *Streptococcus* Group B Type VIII antisera (SSI Diagnostica), a negative reaction was observed for the ΔcpsR mutant and the ΔcpsE acapsular control. We then complemented the ΔcpsR mutant with *cpsR* on a plasmid, and the positive agglutination phenotype returned, matching the wild-type serotype VIII strain (Fig. 4A). In addition, we compared relative amounts of

**TABLE 1** Top BLASTp result for each protein encoded in the variable region of the serotype VIII *cps* locus[a]

| Protein name | Species containing the most similar protein | Amino acid percent identity |
|---|---|---|
| CpsE | *S. suis* and *S. orisratti* | 64.2% |
| CpsR | *S. suis* | 78.0% |
| CpsH | *S. gallolyticus* | 69.4% |
| CpsJ | *S. equinus* and *S. gallolyticus* | 64.6% |

[a]BLASTp results of each of the listed serotype VIII proteins in column one. These proteins were picked because the genes that encode them are most dissimilar in nucleic acid sequence to their cognate genes in the other GBS serotypes. Column 2 reports the species that contained the most similar protein. Column 3 shows the amino acid identity between the serotype VIII protein and the most similar protein by BLASTp aligned along the full length of the protein's amino acid sequence. The top 100 BLASTp hits for each protein, including CpsK, which had no sequence >42% identity, are reported in Table S2.

capsule on our mutants via a whole-cell enzyme-linked immunosorbent assay (ELISA) using a serotype VIII-specific polyclonal antibody. The parent strain had similar levels of capsule to the other serotype VIII clinical isolates tested (Fig. S1). The Δ*cpsR* strain had similar levels of capsule to the acapsular Δ*cpsE* strain, and capsule production was restored in the revertants and the *cpsR*-complemented strain (Fig. 4B). From these results, we concluded that *cpsR* is essential for serotype VIII CPS production.

Although the Δ*cpsR* mutant phenocopied the Δ*cpsE* mutant in the latex agglutination and ELISA experiments, both assays use a serotype VIII-specific antibody. To confirm this phenotype, we used hydrophobicity of the bacterial cell surface as an antibody-independent method to assess encapsulation. CPS is hydrophilic and negatively charged due to the polarity of carbohydrates and the terminal sialic acid present in GBS CPS. Therefore, increased hydrophobicity can be used as a proxy for decreased capsule content. First, we observed that both the Δ*cpsR* and acapsular mutants sedimented in an overnight culture, while the wild-type and revertant strains did not sediment (Fig. 4D). One explanation of sedimentation is the autoaggregation of hydrophobic bacteria separating from the aqueous culture supernatant (40). A hydrophobicity assay using the liquid hydrocarbon, n-hexadecane, was then used to quantify the percentage of bacteria that separated from the aqueous fraction of the solution (41). The Δ*cpsR* mutant phenocopied the acapsular mutant in the hydrophobicity assay, while the revertants phenocopied the wild-type strain (Fig. 4C). Together, the observed increase in hydrophobicity of our mutant suggests that *cpsR* is also required for capsule production, further supporting the hypothesis that acquisition of *cpsR* by GBS was essential for generating this new serotype.

## *S. suis* and *S. gallolyticus cpsF* complement the function of *cpsR* in GBS

Serotype VIII was identified in the human population as far back as the 1970s, but we do not know when it first emerged. Because of this, GBS and the strain that donated its genetic material have both had time to evolve. This makes it difficult, if not impossible, to determine the exact horizontal genetic transfer that created GBS serotype VIII. We conducted functional studies to understand if genetic material from another streptococcal species is sufficient to complement the production of serotype VIII capsule in a *cpsR*-deficient strain of GBS. We hypothesized that the rhamnosyltransferases of closely related streptococcal species could complement the function of CpsR and chose to complement our Δ*cpsR* mutant with *cpsF* from *S. suis* and *S. gallolyticus*. The *S. suis* gene was chosen because the CpsF protein was the closest in the BLASTp results, and the *S. gallolyticus* gene was chosen because the surrounding proteins encoded in the *cps* locus were most similar to those of *S. gallolyticus*. Both complemented mutants had restored agglutination phenotypes and produced capsule (Fig. 4A and B). Therefore, rhamnosyltransferases from other streptococcal species can complement the function of CpsR. This supports our hypothesis that the genetic transfer of *cps* biosynthesis genes likely originated from another streptococcal species.

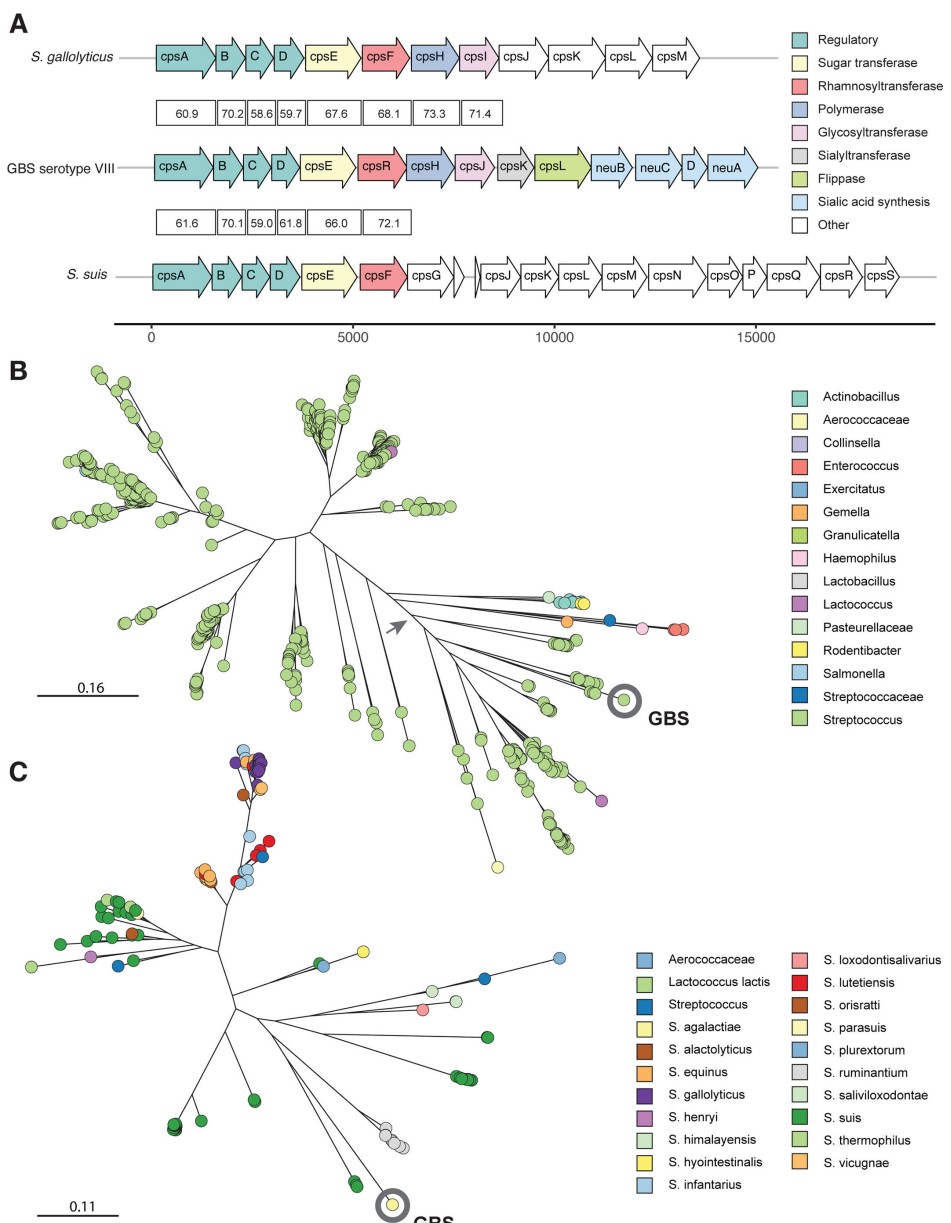

**FIG 3** Relatedness of serotype VIII *cpsR* and the encoded rhamnosyltransferase to other streptococcal species. (A) Boxes between arrows show nucleotide identity of the *cps* locus genes. Strains of *S. suis* (GenBank accession number AB737838) and *S. gallolyticus* (GenBank accession number FR824043) are both being compared to the center strain GBS serotype VIII AR977. Color fill of arrows represents conserved function as in Fig. 1. (B) Maximum-likelihood phylogenetic tree of CpsR amino acid sequences. The tree is drawn to scale, and branch lengths correspond to the number of substitutions per site. Colors represent genus. Arrow indicates the node that was selected to create the zoomed-in tree in panel C. (C) Subtree of the most closely related sequences to GBS CpsR. Colors correspond to species.

## Wild-type serotype VIII outcompetes ΔcpsR in a murine model of vaginal co-colonization

Capsule aids GBS in successfully colonizing and competing in the vaginal tract (14). Because our Δ*cpsR* mutant is deficient in producing serotype VIII capsule, we aimed to assess the role of *cpsR* in vaginal colonization. We performed a murine vaginal co-colonization model using wild-type serotype VIII strain AR977 and our Δ*cpsR* mutant. On day 1 post-colonization, the competitive index comparing wild-type to Δ*cpsR* was 4, indicating

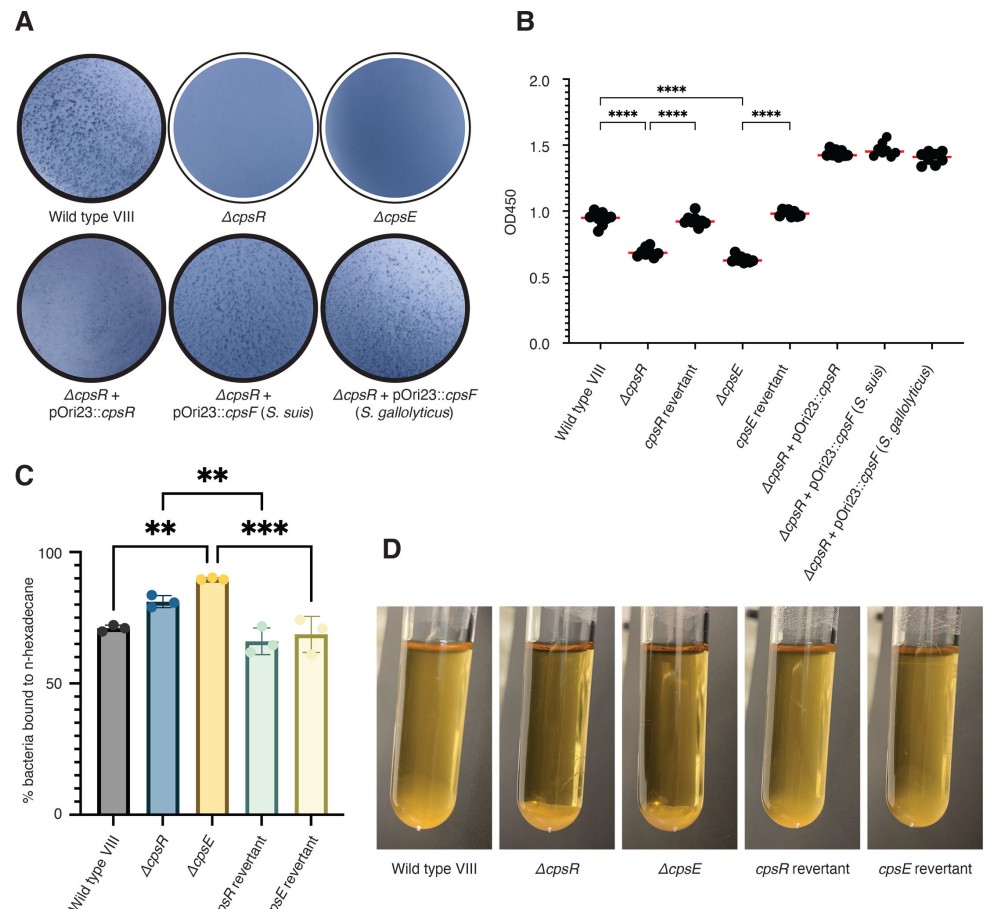

**FIG 4** *cpsR* is required for serotype VIII capsule production and can be complemented by *cpsF* from other streptococcal species. (A) Latex agglutination performed using serotype VIII-specific antisera. A positive reaction is indicated by dark blue dots in the liquid (black outline), while negative reactions appear clear (white outline). (B) Whole-cell ELISA using serotype VIII-specific primary antibody to assess relative levels of total capsule. Dots represent technical replicates (*n* = 8), and the red bar represents the median. Significance was determined using Tukey's multiple comparisons test (****P < 0.0001). (C) Hydrophobicity assay. Significance was determined using Tukey's multiple comparisons test (**P < 0.01, ***P < 0.001). (D) Photos of sedimentation phenotype after overnight growth.

that although both strains were recovered, the wild-type strain had an initial advantage (Fig. 5). By day 5, all of the mice were colonized with only the wild-type VIII strain, resulting in the maximum competitive index of 361 (based on a maximum colony count of 200 and limit of detection of 0.5). This suggests GBS serotype VIII strains that contain a functional *cpsR* have a competitive advantage in *in vivo* colonization.

## DISCUSSION

Serotype replacement has the potential to occur via three mechanisms: expansion of non-vaccine-type serotypes, intraspecies capsule switching, or generation of new serotypes. Serotype VIII is potentially relevant to all three examples. First, already circulating serotype VIII strains could replace vaccine strains in colonization and disease, as has been seen with the introduction of the pneumococcal conjugate vaccine (42–44). Second, serotype VIII *cps* genes could be transferred to vaccine serotypes and allow those newly encapsulated strains to escape vaccine protection. Intraspecies capsule switching in GBS has already been documented (45–47). And third, serotype VIII is a demonstration that HGT from other species can give rise to a completely new GBS serotype. Interspecies capsule switching has been reported between pneumococcus and

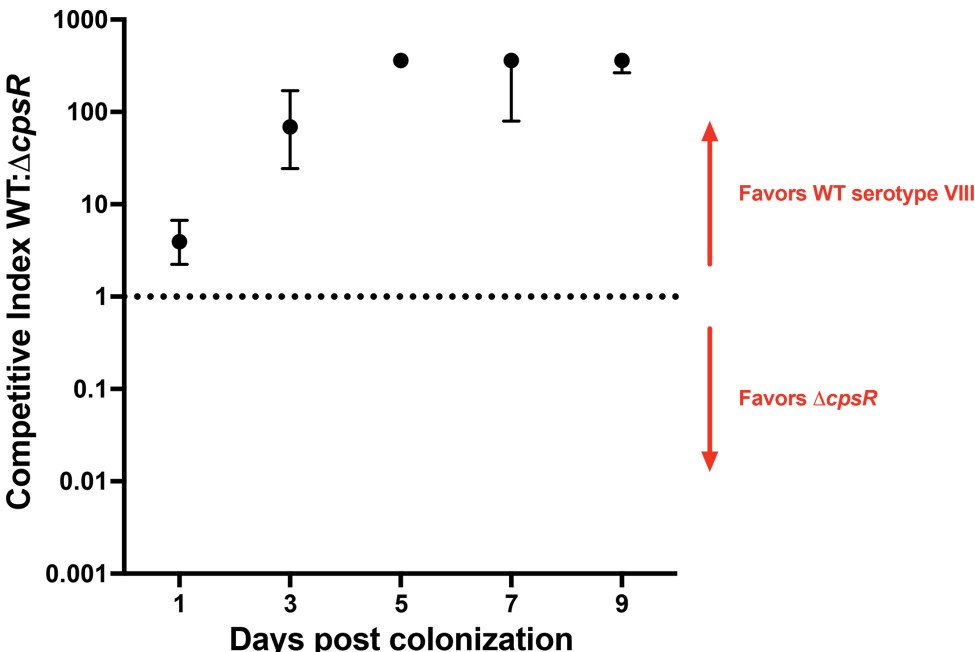

FIG 5 Wild-type (WT) serotype VIII outcompetes the isogenic mutant ΔcpsR in a murine model of vaginal co-colonization. Adult non-pregnant C57BL/6J mice were vaginally colonized with equal amounts of WT VIII and ΔcpsR (n = 8). Samples were collected by vaginally swabbing the mice on days 1, 3, 5, 7, and 9 post-colonization. Swab contents were plated on CHROMagar; WT VIII and ΔcpsR colonies were differentiated by immunoblotting with a serotype VIII-specific polyclonal antibody. Representative positive and negative colonies were verified for each time point via PCR. Data points represent median competition indices, and error bars represent 95% confidence intervals.

other streptococcal species (48, 49). The unique structure of serotype VIII among GBS serotypes indicates its likely evolution from interspecies HGT.

Cieslewicz et al. proposed two groups of interrelated GBS serotype capsular polysaccharide structures (37). One group included serotypes Ia, Ib, III, and VI, which differ by single linkage or monosaccharide substitutions. This group also contains serotype II, which they report to be most closely related to serotype III but with two additional monosaccharides. The other group included serotypes IV, V, and VII, which differ only in the location of the side-chain linkage to the trisaccharide backbone or the addition of a monosaccharide side chain for serotype V. Serotype IX was later shown to be related to V and VII and should also be included in this group (50). Serotype VIII did not fit into either of these interrelated groups because of the rhamnose in its trisaccharide backbone. This suggests that, while the other GBS serotypes are genetically related and some were created by intraspecies HGT, serotype VIII is genetically unique, and the addition of a rhamnosyltransferase may have arisen from an interspecies genetic transfer event.

In this work, we have shown that cpsR is essential for the production of wild-type serotype VIII capsule. CpsR is the rhamnosyltransferase responsible for the addition of the unique rhamnose in the serotype VIII CPS. When we made an in-frame deletion of cpsR, the bacteria no longer reacted with the serotype VIII latex agglutination antibody. In addition, the ΔcpsR mutant phenotypically copied the acapsular control, ΔcpsE, in a whole-cell ELISA. To investigate the functionality of potentially transferred genetic material, we complemented our ΔcpsR mutant with cognate rhamnosyltransferases from closely related streptococcal species. *S. suis* and *S. gallolyticus cpsF* genes were individually introduced to our ΔcpsR mutant. Both complemented strains restored capsule production and recognition of the bacteria by the serotype VIII latex agglutination antibody. In addition, we found cpsR to be essential to compete with wild-type serotype VIII in a murine model of vaginal co-colonization. Our results show the potential for

interspecies genetic transfer of *cps* genes to create functional capsular polysaccharide structures which could confer a competitive advantage in *in vivo* colonization.

Additional functional evidence for GBS using *cps* genes from other streptococcal species was shown by Roy et al. (51). GBS and *S. suis* are the only known gram-positive organisms with sialylated CPS, but they utilize different linkages. When the GBS α–2,3-sialyltransferase was swapped with the *S. suis* α-2,6-sialyltransferase in serotype III and V strains, newly encapsulated mutants with an α-2,6 sialic acid linkage were constructed. When the GBS α-2,3-sialyltransferase was introduced to *S. suis*, unencapsulated mutants were produced. They hypothesized that the CPS biosynthesis machinery in GBS is more permissive and therefore was able to use the sialyltransferase from *S. suis* in capsule construction (51). This further demonstrates the permissiveness of the GBS *cps* loci to use foreign DNA to construct uniquely encapsulated GBS.

*S. suis* and *S. gallolyticus* have a natural opportunity for HGT with GBS, as they have overlapping niches. *S. suis* is primarily a porcine pathogen colonizing the upper respiratory, gastrointestinal, and genital tracts of pigs and causes invasive infection, such as meningitis and septicemia (52, 53). It is also a zoonotic pathogen, causing invasive infection in humans when contaminated undercooked pork is consumed (54). Outbreaks in human populations are most common in Southeast Asia (55). *S. gallolyticus* is part of the *Streptococcus bovis/Streptococcus equinus* complex (SBSEC). It is a commensal species of bovine and equine gastrointestinal tracts (56). In humans, *S. gallolyticus* causes infective endocarditis and promotes colorectal cancer (57, 58). In addition to both GBS and *S. gallolyticus* existing in human hosts, GBS also causes disease in cows and was originally identified from the milk of a cow with bovine mastitis (59).

GBS is a multi-host bacterial pathogen infecting fish, frogs, and camels in addition to humans and cows (60). Given its broader host range, GBS has the opportunity to overlap with many streptococcal species in addition to the two we tested. This includes many other species from our BLASTp results, particularly other multi-host bacterial pathogens such as the other SBSEC species or *Streptococcus ruminantium*. *S. ruminantium* was in close proximity to GBS on our CpsR phylogenetic tree but did not show up in our BLASTp results, most likely because it was recently reclassified from *S. suis* (61). Additionally, lineages of GBS that can infect multiple hosts have higher genome plasticity when compared to host specialist lineages of GBS (60). This suggests that GBS strains capable of colonizing multiple hosts have an increased likelihood of HGT and may have the potential to create new serotypes.

GBS is not known to be naturally competent, so intraspecies genetic recombination presumably occurs via conjugation (62). GBS has a higher rate of genome-wide recombination to mutation in comparison to other streptococcal species, and its genome includes >100 kb regions of recombination that contain the *cps* locus (62, 63). HGT of capsule biosynthesis genes has already been documented among GBS isolates. A serotype III-to-IV switch was discovered in which a 35 kb fragment of DNA, including the entire *cps* locus, was transferred from a serotype IV strain into a clonal complex 17 strain (45). CC17 strains were previously all typed as serotype III. This could be a vehicle for further capsule switching or novel capsule creation. In addition, conjugation-based genetic transfer between *S. suis* and GBS has been demonstrated (64). The integrative conjugative element ICE*su*32457 in *S. suis* has a ~1.3 kb region conserved in the GBS ICE*Sa*2603. In mating experiments, *S. suis* was able to transfer its ICE to GBS, creating a hybrid conjugative element (65). The authors note that this demonstrates a potential mechanism for genetic transfer between animal and human streptococci.

Functional work can quickly become relevant, as seen in pneumococcus. In work published in 2020, Nahm et al. showed that giving acapsular, nonvirulent pneumococcus the *cps* locus from an oral streptococcal species made the mutant virulent in a mouse model (66). They highlighted the potential of novel encapsulation of pneumococcus via HGT from oral streptococcal species. Pneumococcus and oral streptococcal species share niches, and oral streptococcal strains already act as a genetic reservoir for antibiotic resistance for pneumococcus, but they note *cps* transfer had not yet been observed in

nature (67). Importantly, though, immunogenic assays and sequencing of *cps* loci had demonstrated identical capsule expression between pneumococcus and *Streptococcus mitis,* suggesting previous transfer of *cps* genes from *S. mitis* to pneumococcus (49, 68). This work proved to be immediately relevant as a new serotype of *S. pneumoniae* was soon discovered that was likely the result of genetic transfer from the oral strain *S. mitis.* A 6 kb fragment of the new serotype's *cps* locus had 94% amino acid identity with *S. mitis* (48). This transfer created a new serotype in a human pathogen.

In conclusion, it is likely that interspecies genetic transfer created GBS serotype VIII. This mechanism of generating capsule diversity could produce novel GBS serotypes under the pressure of CPS-conjugate vaccine selection, with other streptococci acting as a genetic reservoir for GBS.

## MATERIALS AND METHODS

### Phylogenetic tree

One hundred CpsR amino acid sequences captured by BLASTp, aligning the CpsR sequence from wild-type serotype VIII AR977 to the NCBI's nr database, were aligned using MUSCLE v.3.7 (69) with default parameters, except that the output format was set to "-clw" (ClustalW) for compatibility as input for HMMER v.3.3 tools (70). An HMM model was then built using this alignment with hmmbuild, and the HMM was searched against NCBI's nr database (downloaded on 30 August 2024) with hmmsearch, both with default parameters. Eight hundred twenty sequences were captured by parsing the hmmsearch output for a bit score >700. These 820 sequences, together with the AR977 canonical serotype VIII CpsR amino acid sequence, were aligned with MUSCLE (all default), and the resulting alignment in fasta format was used as input for megacc v.10.0.5 (71) to generate a maximum-likelihood phylogenetic tree of CpsR amino acid sequences.

The evolutionary history was inferred by using the maximum-likelihood method andjones-taylor-thornton (JTT) matrix-based model (72). The tree with the highest log likelihood ($-42{,}184.15$) is shown. Initial tree(s) for the heuristic search were obtained automatically by applying neighbor-join and BioNJ algorithms to a matrix of pairwise distances estimated using the JTT model, and then selecting the topology with the superior log likelihood value. The tree is drawn to scale, with branch lengths measured in the number of substitutions per site. This analysis involved 821 amino acid sequences. There were a total of 556 positions in the final data set. Evolutionary analyses were conducted in MEGA11 (73, 74).

### Bacterial strains and growth conditions

Bacterial strains used in this work are listed in Table 2. GBS serotype VIII clinical isolates were obtained from the CDC's ABCs. All genetic manipulation was performed in the AR977 genetic background. The clinical isolates, in-frame deletion mutants, and revertants were all grown stationary at 37°C in tryptic soy broth. Complementation mutants were grown stationary at 37°C in tryptic soy broth with the addition of 5 µg/mL of erythromycin for plasmid maintenance. *Escherichia coli* was grown shaking in Luria-Bertani broth with the addition of 300 µg/mL of erythromycin to maintain pMBSacB (28°C) or pOri23 (37°C).

### Generation of GBS serotype VIII mutant strains

In-frame deletion mutants of *cpsR* and *cpsE* were created by allelic exchange using a temperature-sensitive shuttle vector with a counterselectable sucrose sensitivity cassette, pMBSacB (75). Eight hundred base pairs of homology upstream and downstream of the target gene were cloned into pMBSacB, cut with XhoI and NotI, via Gibson assembly, and transformed into 5-alpha competent *E. coli* (New England Biolabs). Because the end of the coding sequence for *cpsR* overlaps the beginning of *cpsH*, instead of deleting the entire gene from start codon to stop codon, the downstream region of

**TABLE 2** GBS strains and plasmids used in this study

| Strain | Description | Source/CDC strain #/ Reference |
|---|---|---|
| 5-Alpha competent *E. coli* | Chemically competent cloning strain | New England Biolabs item no. C2987H |
| AR977 | Serotype VIII used as the genetic background for all mutagenesis work in this study | CDC strain 2013226269 |
| Δ*cpsR* | AR977 with an in-frame deletion of *cpsR* from start codon of *cpsR* to start codon of *cpsH* | This study |
| *cpsR* revertant | AR977 revertant control for allelic exchange mutagenesis of Δ*cpsR* | This study |
| Δ*cpsE* | AR977 with an in-frame deletion of *cpsE* from start to stop codon of *cpsE* | This study |
| *cpsE* revertant | AR977 revertant control for allelic exchange mutagenesis of Δ*cpsE* | This study |
| Δ*cpsR* pOri23::*cpsR* | Δ*cpsR* complemented with the shuttle vector pOri23 containing *cpsR* under the control of the *p23* promoter | This study |
| Δ*cpsR* pOri23::*cpsF* (from *S. suis*) | Δ*cpsR* complemented with the shuttle vector pOri23 containing *cpsF* under the control of the *p23* promoter | This study |
| Δ*cpsR* pOri23::*cpsF* (from *S. gallolyticus*) | Δ*cpsR* complemented with the shuttle vector pOri23 containing *cpsF* under the control of the *p23* promoter | This study |
| pMBSacB | Temperature-sensitive, erythromycin-resistant plasmid containing p23-*sacB* for sucrose-counterselectable GBS mutagenesis | (75) |
| pOri23 | Shuttle vector containing the p23 promoter upstream of a multicloning site; erythromycin-resistant | (76) |

homology began at the start codon for *cpsH* to preserve the expression of *cpsH* and downstream genes. The resulting plasmids were transformed into an electrocompetent serotype VIII clinical isolate, AR977. Single-cross plasmid insertions were selected for by growing the strains with erythromycin at 37°C. Then double-cross plasmid excisions were selected for by growing strains at 28°C with no antibiotic selection. Lastly, strains were grown in sucrose to select for double-crosses that had completely removed the plasmid. Deletion mutants and revertants were confirmed via PCR and sequencing of the *cps* locus.

Complementation of Δ*cpsR* was performed by cloning *cpsR* from GBS and *cpsF* from both *S. gallolyticus* and *S. suis* into the expression vector pOri23 cut with BamHI and PstI (76). *cpsR* was PCR amplified from the parent serotype VIII clinical isolate, AR977, and Gibson-assembled into pOri23 under the control of the *p23* promoter. *cpsF* from *S. gallolyticus* and *S. suis* was ordered as gene blocks from Integrated DNA Technologies. PCR was performed on the gene blocks to enable Gibson assembly of each gene into pOri23. Complementation plasmids were first transformed into 5-alpha competent *E. coli* (New England Biolabs) and then into electrocompetent Δ*cpsR*. Successful transformants were selected using erythromycin. All primer and gene block sequences used are listed in Table S3.

## Latex agglutination

Immulex *Streptococcus* Group B Type VIII antisera (SSI Diagnostica) was used to detect production of serotype VIII capsule. Individual colonies were picked and mixed with 10 µL of water on a reaction card. One drop of Immulex type VIII antisera was added, and the solution was mixed for 30 seconds. A positive reaction was recorded if agglutination could be visualized in the mixture.

## Enzyme-linked immunosorbent assay

A whole-cell ELISA was used to compare relative amounts of capsule present on each strain. A serotype VIII-specific polyclonal antibody was used to detect capsule (SSI Diagnostica). A previously described protocol was optimized for detecting serotype VIII capsule (14). GBS strains were grown overnight in tryptic soy broth, stationary at 37°C. In the morning, strains were sub-cultured and grown to an $OD_{600}$ of 0.6. Cultures were then centrifuged, and the pellet was resuspended in the same volume of coating buffer (0.087 M $NaHCO_3$ and 0.015 M $Na_2CO_3$ in sterile water, pH 9.5). Suspensions were diluted 1:10 in

coating buffer, and 100 µL of each strain was added to the wells in a column of an ELISA plate. The plate was incubated overnight at 37°C.

The following day, any liquid left in the wells was removed, and wells were fixed with 50 µL of 100% methanol for 20 minutes at 37°C. Excess methanol was removed, and wells were left to dry for another 20 minutes. Wells were then washed with 200 µL of phosphate-buffered saline (PBS) for 15 minutes at room temperature. Samples were blocked with 200 µL of 1% bovine serum albumin (BSA) in PBS for 4 hours, covered, at 37°C. After blocking, 50 µL of serotype VIII-specific antibody diluted 1:15,000 in PBS with 1% BSA, 0.05% Tween 20 (PBSAT) was added to the wells and incubated for 1 hour, covered, at room temperature. Wells were subsequently washed three times with 200 µL of PBSAT for 5 minutes each. Secondary antibody, goat anti-rabbit IgG-horseradish peroxidase (HRP) (Invitrogen), was diluted 1:15,000 in PBSAT, and 50 µL was added to each well to incubate for 15 minutes at 37°C. Wells were again washed three times with 200 µL of PBSAT for 5 minutes each. One hundred microliters of 3,3',5,5'-tetramethylbenzidine (TMB)-ELISA substrate solution (Pierce) was added to each well and incubated for 10 minutes at room temperature away from light. The reaction was stopped by adding 50 µL of 1N $H_2SO_4$. Absorbance was read at 450 nm in a plate reader.

## Hydrophobicity assay

This hydrophobicity assay was adapted from Campeau et al. (77). Two milliliters of overnight culture was pelleted via centrifugation. Pellets were washed twice with sterile PBS and then resuspended in 2 mL of sterile PBS. Resuspensions were transferred to borosilicate test tubes. Four hundred microliters of n-hexadecane was pipetted on top of the resuspended pellet. Negative controls containing only PBS and pellet were prepared in parallel. All tubes were then covered and vortexed for 15 seconds. Tubes were then rested for 5 minutes to allow the aqueous and organic layers to separate. The aqueous layer was removed with a pipette, and the $OD_{600}$ was measured. Hydrophobicity was assessed as

$$100 - \left( \frac{OD_{600} \text{ sample}}{OD_{600} \text{ negative control}} \right) \times 100 = \% \text{ bacteria bound to n-hexadecane}$$

## Murine vaginal co-colonization model

Six- to 8-week-old female C57BL/6J mice were purchased from Jackson Laboratories. As previously described (14, 78), the animals' estrous cycles were synchronized by subcutaneously injecting 10 µg of water-soluble 17β-estradiol (Sigma-Aldrich), 48 and 24 hours prior to colonization. Wild-type serotype VIII and Δ$cpsR$ strains were grown overnight. In the morning, the optical densities of the two cultures were normalized to each other. Cultures were then centrifuged and resuspended in a 1:1 mixture of the two strains in PBS, which was subsequently mixed 1:1 with sterile 10% gelatin. The final concentration was $1 \times 10^9$ colony-forming units (CFU)/mL. Mice were colonized under anesthesia with 3%-5% isoflurane (Baxter). Fifty microliters of the GBS mixture (total inoculum $5 \times 10^7$ CFU) was administered intravaginally using a sterile pipette. Post-inoculation, mice were housed separately and swabbed on days 1, 3, 5, 7, and 9 post-colonization. Vaginal swabs were vigorously stirred in 300 µL of sterile PBS. The swab solution was serially diluted and plated for CFU enumeration. The dilutions were also spread on CHROMagar plates (CHROMagar) to generate single colonies and to identify GBS from other vaginal microbes. Colony immunoblots were performed on these plates using serotype VIII-specific polyclonal antibody (SSI Diagnostica) to differentiate wild-type VIII from Δ$cpsR$ colonies and calculate competitive index.

## Colony immunoblot

For each mouse at each time point, a dilution plate with 20-200 colonies was selected for colony immunoblot. Nitrocellulose membranes were applied to each plate and lifted to

transfer colony contents. Membranes were blocked in 3% BSA in PBS (blocking solution) overnight at 4°C. Blots were transferred to blocking solution containing *Streptococcus* Group B type VIII serum (SSI Diagnostica) diluted 1:20,000 and incubated for 30 minutes, shaking, at room temperature. Blots were then washed three times in PBS, shaking, at room temperature for 10 minutes per wash. Blots were transferred to a blocking solution containing the secondary antibody goat anti-rabbit IgG-HRP (Invitrogen) diluted 1:1,000 for 2 hours with shaking at room temperature. Blots were again washed three times with PBS and then stained using a 3,3′-diaminobenzidine tetrahydrochloride substrate kit (Abcam). Positive and negative colonies were counted to calculate the ratio of wild-type VIII to Δ*cpsR* colonization.

Competitive index was calculated as

$$\frac{\text{CFU wild} - \text{type recovered } \div \text{ CFU wild} - \text{type inoculated}}{\text{CFU } \Delta cpsR \text{ recovered } \div \text{ CFU } \Delta cpsR \text{ inoculated}} = \text{competitive index}$$

## Statistical analyses

Statistics were calculated using GraphPad Prism 9 Software. ELISA absorbance and hydrophobicity between strains were compared using a one-way analysis of variance with Tukey's multiple comparisons test. Significance was denoted as *$P < 0.05$, **$P < 0.01$, ***$P < 0.001$, and ****$P < 0.0001$.

## ACKNOWLEDGMENTS

This work was supported by NIH/NIAID award R01AI155476 (to A.J.R.). M.E.S. received support from NIH/NIAID training grant T32AI007180.

## AUTHOR AFFILIATIONS

[1]Department of Microbiology, New York University Grossman School of Medicine, New York, New York, USA

[2]Department of Pediatrics, New York University Grossman School of Medicine, New York, New York, USA

[3]Department of Microbiology and Immunology, Institute for Genome Sciences, University of Maryland School of Medicine, Baltimore, Maryland, USA

## AUTHOR ORCIDs

Molly E. Sharp  http://orcid.org/0000-0002-2551-2614
Adam J. Ratner  http://orcid.org/0000-0003-1761-794X

## FUNDING

| Funder | Grant(s) | Author(s) |
| --- | --- | --- |
| National Institute of Allergy and Infectious Diseases | R01AI155476 | Adam J. Ratner |
| National Institute of Allergy and Infectious Diseases | T32AI007180 | Molly E. Sharp |

## AUTHOR CONTRIBUTIONS

Molly E. Sharp, Conceptualization, Data curation, Formal analysis, Investigation, Methodology, Validation, Visualization, Writing – original draft, Writing – review and editing | Julia Sproch, Investigation | Sydney Haldeman, Formal analysis | Hervé Tettelin, Data curation, Formal analysis, Methodology, Software, Writing – review and editing | Adam J. Ratner, Conceptualization, Funding acquisition, Methodology, Project administration, Resources, Supervision, Visualization, Writing – review and editing

## DATA AVAILABILITY

The whole genome sequence for strain AR977 is available at the National Center for Biotechnology Information BioProject accession number PRJNA986900.

## ETHICS APPROVAL

All experimental procedures were reviewed and approved by the NYU Langone Institutional Animal Care and Use Committee.

## ADDITIONAL FILES

The following material is available online.

### Supplemental Material

**Fig. S1 (Spectrum01227-25-s0001.docx).** ELISA data of wild-type VIII strains.
**Supplemental tables (Spectrum01227-25-s0002.docx).** Tables S1 and S3.
**Table S2 (Spectrum01227-25-s0003.xlsx).** Top 100 BLASTp hits.

### Open Peer Review

**PEER REVIEW HISTORY (review-history.pdf).** An accounting of the reviewer comments and feedback.

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
