## [Reviewer comments · Microbiology Spectrum]

Microbiology Spectrum

Expansion of the Group B *Streptococcus* serotype repertoire via gene acquisition from other streptococcal species

Molly Sharp, Julia Sproch, Sydney Haldeman, Herve Tettelin, and Adam Ratner

Corresponding Author(s): Adam Ratner, New York University Grossman School of Medicine

Review Timeline:

Submission Date:	April 23, 2025
Editorial Decision:	July 6, 2025
Revision Received:	August 12, 2025
Accepted:	September 8, 2025

Editor: Nicolas Soler

Reviewer(s): Disclosure of reviewer identity is with reference to reviewer comments included in decision letter(s). The following individuals involved in review of your submission have agreed to reveal their identity: Ryan S Doster (Reviewer #1)

Transaction Report:

DOI: <https://doi.org/10.1128/spectrum.01227-25>

Re: Spectrum01227-25 (Expansion of the Group B *Streptococcus* serotype repertoire via gene acquisition from agricultural streptococci)

Dear Dr. Adam J. Ratner:

Thank you for the privilege of reviewing your work. Below you will find my comments, instructions from the Spectrum editorial office, and the reviewer comments.

The reviewers all agreed that the manuscript was clear and the conclusions supported by the robust results. However they asked several modifications that I ask you to answer point by point. Moreover, please correct the focus on the writing which was focuses mainly on the comparison with agricultural streptococci, and avoid cherry-pick from the results.

Revision Guidelines

Sincerely,
Nicolas Soler
Editor
Microbiology Spectrum

Reviewer #1 (Comments for the Author):

This manuscript by Sharp et al. examines the origin of GBS serotype VIII strains. These strains express cpsR, a rhamnosyltransferase, which is unique to serotype VIII strains. Homologous genes to CpsR were found in closely related species

including *S. suis* and *S. galloyticus*. A *cpsR* mutant no longer expressed the type VIII capsule as denoted by VIII specific antibodies binding and ELISA, and like other GBS capsule knockout mutants, mutants lacking *cpsR* showed impaired vaginal colonization compared to the parent strain. Complementing *cpsR* from other *Streptococcus* species restored function, demonstrating that GBS serotypes VIII strains likely acquired this gene (and the *cps* locus) via horizontal gene transfer along, which may be of importance as the field moves closer to a GBS vaccine, which only contains antigens for GBS serotypes Ia through V.

Overall, this manuscript is well written with a clear progression from the sequence homology to *cpsR* mutant experiments demonstrating the relevance of the gene in GBS serotype VIII capsule and vaginal colonization. While these strains still represent a minority of those causing clinical disease, the novel evolutionary nature of the serotype VIII strains with respect to vaccine development is interesting. Overall, the methods, data presented, and conclusions are well aligned. While the scope of this manuscript is fairly narrow, the studies are well done. This Reviewer has no major critiques to this work, but a couple minor critiques requiring text changes are noted below.

Minor critiques:

1. In 2C *S. ruminantium* appears have a gene closely related to GBS *CpsR* but did not appear in Sup Table 2. It looks like this species was reclassified at some point from *S. suis* serotype 33. I am not sure if it is possible to determine which *S. suis* sequences in Sup Table 2 would now be classified as *S. ruminantium*.
2. In Lines 192-198 it appears that the autoaggregation/sedimentation (D) and hydrophobicity (C) are switched in Figure 3 compared to the text.
3. It appears that both the capsule ELISA and hydrophobicity data has relatively high background (*cpsR* and *cpsE* mutant levels compared to the wild type and revertant), is that common for these assays and is there an explanation (ie non-specific binding of the anti-VIII antibody)? Would you expect a GBS strain of another serotype to have similar ELISA readings compared to the *cpsR* or E mutant?

Reviewer #2 (Comments for the Author):

General comment

This is an interesting, important and well-executed study that highlights the potential for "vaccine escape mutants" to arise in the multi-host pathogen *Streptococcus agalactiae* or group B *Streptococcus* (GBS). The evolutionary evidence for the possibility of inter-species transfer and the functional evidence for the role of serotype VIII are convincing. The main flaw of the current write-up is that the authors cherry-pick from their results, apparently to align with their chosen focus on comparison with agricultural streptococci. Results, discussion and title need to be modified to accurately reflect results shown in Table S2, as detailed below.

The abstract is clear and largely well-supported by data, with the exception of line 38. This line is incorrect in two ways, i.e., the species listed are not the closest matches (other species are equally as close based on Table S2 but not mentioned) and the niche of *S. suis* (mostly found in pigs) doesn't overlap with that of GBS (rarely found in pigs) unless the niche overlap occurs in humans infected with *S. suis*, which is a possibility (e.g., in China, where human *S. suis* occurs more commonly than anywhere else).

Importance section: Clear and informative, although comments about colonization, pathogenicity and vaccine candidates should be qualified as relating to human GBS. In bovine and piscine GBS, pseudogenisation of the capsular operon is observed (e.g., Almeida, *Environ Microbiol.* 2016 Nov;18(11):4216-4229. doi: 10.1111/1462-2920.13550) implying a lack of functional relevance, and GBS vaccines in fish and cattle are not based on CPS.

The introduction is very clear and very well-referenced.

Genomic and laboratory-based methods are informative and elegant.

The results section is very accessible and nicely illustrated but appears to lack objectivity. In Table 2, authors cherry-picked from the results presented in Table S2. For *cpsE*, scores for *S. orisratti* and *S. suis* were identical yet *S. orisratti* is not mentioned, maybe because it isn't an "agricultural streptococcus"? Likewise, for *cpsJ*, scores are identical for *S. equinus* and *S. galloyticus* yet only *S. equinus* is listed in Table 1. Indeed, authors state (line 205-206) that donor and recipient have had time to evolve since a historical genetic transfer. Why, then, do authors not consider other potential donors that are close matches, such as *S. thermophilus* (*CpsR*), *S. lutetientis* (*CpsH*) or *S. infantarius* (*CpsJ*)? The entire manuscript, including Table 2, results as presented in line 133-140, Figure 2A, and all related discussion as well as the manuscript title and abstract need to be updated to reflect results from Table S2, which do not support the association with agricultural streptococci as *Streptococcus orisratti* is not an agriculture-related species and neither is *Strep. galloyticus*. While the latter has certainly been reported from agricultural animals, e.g. turkeys and dairy cattle, it is more commonly associated with humans, particularly with colorectal cancer (e.g. *Gut Microbiome* (Camb). 2024 Nov 4;5:e9. doi: 10.1017/gmb.2024.11.) and invasive disease in infants (e.g., *Pediatr Infect Dis J.* 2022 Nov 1;41(11):e494-e497. doi: 10.1097/INF.0000000000003682), putting it in the same niches as GBS in humans. While the selective presentation of results does not affect the core premise of the manuscript, i.e. that interspecies transfer may contribute to evolution of GBS capsular serotypes with functional significance, it does affect the surrounding narrative, e.g. the

title and abstract (as already stated) and the conclusion drawn in line 141-142.

From Figure 2C, it is difficult to ascertain whether the GBS sequence is more closely related to *S. suis* or *S. ruminantium*. The branch lengths of the tree rather than the positioning of the coloured leaves of the tree are the salient information here and insufficient detail about branch lengths is provided to assess which one is closer. The close proximity of *S. gallolyticus* to GBS CpsR, as described in line 162-164, is not supported by Figure 2C.

In Figure 3, panels C and D appear to have been inverted. Text for figure 3C (line 193) refers to the image in panel D, and text for figure 3D (line 198) refers to the image currently presented in panel C.

In lines 198 and 265, authors state that the cpsR mutant phenocopied the acapsular strain. Figure 3, however, appears to show non-overlapping error bars for those two mutants and a P-value is, intentionally or unintentionally, withheld, so while resemblance is demonstrated, with the data provided, phenocopying is not.

Line 208 asserts that authors wish to understand the potential for genetic transfer. This is not assessed in the current study. Rather, authors assess whether the phenotype of one species can be induced with genetic material from another species in an in vitro system. Whilst I don't dispute the evidence they provide in support of this in vitro phenomenon, nor the hypothesis relating to HGT (line 217-218), genetic transfer itself is not tested. Differences in HGT may exist between and within gram-positive bacterial species and the potential for HGT between GBS and the suggested species, or other species that may be equally as relevant based on Table 2S, is not assessed. Likewise, creation of mutants and revertants and demonstration of their functional impacts is no evidence of an evolutionary relationship (line 266).

The discussion opens with three mechanisms for serotype replacement (lines 236-238). Could a reference be provided for each of those mechanisms? Please also add a numbered reference for Roy et al. (line 276). Authors cite Roy et al. as stating "Another striking feature of these two pathogens is that *S. suis* and GBS are the only Gram-positive bacteria expressing a sialylated CPS." This is indeed stated in Roy et al., but it is not supported by references in their paper and it is not clear whether all other potential donors (e.g. *S. equinus*, *S. gallolyticus*, *S. infantarius*, *S. orisratti* or *S. thermophilus*, based on Table S2) have been considered. Would the authors be able to comment on whether there is absence of evidence for those species, or evidence of absence?

In l. 286, authors appear to conflate potential (ability to do something, which would be a trait of the bacteria) with opportunity (circumstances allowing for the activity, such as a shared niche). Opportunity for genetic transfer seems most likely in the human gut, where GBS and SBSEC co-exist.

L. 294-296 GBS also cause disease in fishes, camels, frogs and many other species, including farmed and wild species. Why selectively cite its occurrence in dairy cattle?

The discussion lacks consideration of potentially relevant streptococcal species other than *S. suis* and *S. gallolyticus* as listed in table S2. Several of those species, certainly members of the SBSEC, are multi-host pathogens rather than "animal streptococci" as suggested in line 328. Please amend the discussion accordingly.

Reviewer #1 (Comments for the Author):

This manuscript by Sharp et al. examines the origin of GBS serotype VIII strains. These strains express *cpsR*, a rhamnosyltransferase, which is unique to serotype VIII strains. Homologous genes to *CpsR* were found in closely related species including *S. suis* and *S. galloyticus*. A *cpsR* mutant no longer expressed the type VIII capsule as denoted by VIII specific antibodies binding and ELISA, and like other GBS capsule knockout mutants, mutants lacking *cpsR* showed impaired vaginal colonization compared to the parent strain. Complementing *cpsR* from other *Streptococcus* species restored function, demonstrating that GBS serotypes VIII strains likely acquired this gene (and the *cps* locus) via horizontal gene transfer along, which may be of importance as the field moves closer to a GBS vaccine, which only contains antigens for GBS serotypes Ia through V.

Overall, this manuscript is well written with a clear progression from the sequence homology to *cpsR* mutant experiments demonstrating the relevance of the gene in GBS serotype VIII capsule and vaginal colonization. While these strains still represent a minority of those causing clinical disease, the novel evolutionary nature of the serotype VIII strains with respect to vaccine development is interesting. Overall, the methods, data presented, and conclusions are well aligned. While the scope of this manuscript is fairly narrow, the studies are well done. This Reviewer has no major critiques to this work, but a couple minor critiques requiring text changes are noted below.

Thank you for the thoughtful feedback. We appreciate you catching the switched in-text citation of figures and your engagement with the specifics of the ELISA and hydrophobicity assays. Each critique is addressed below.

Minor critiques:

1. In 2C *S. ruminantium* appears have a gene closely related to GBS *CpsR* but did not appear in Sup Table 2. It looks like this species was reclassified at some point from *S. suis* serotype 33. I am not sure if it is possible to determine which *S. suis* sequences in Sup Table 2 would now be classified as *S. ruminantium*.

We agree it would be interesting to understand which of the *S. suis* strains from our initial BLASTp results may be reclassified as *S. ruminantium*. Unfortunately, we do not have adequate information to reclassify existing sequences. We do however think this analysis of the Fig 2C tree was missing from our paper and have added to the discussion to acknowledge this recent reclassification (lines 341-343).

2. In Lines 192-198 it appears that the autoaggregation/sedimentation (D) and hydrophobicity (C) are switched in Figure 3 compared to the text.

Thank you for catching this, the notation in the text has been fixed.

3. It appears that both the capsule ELISA and hydrophobicity data has relatively high background (cpsR and cpsE mutant levels compared to the wild type and revertant), is that common for these assays and is there an explanation (ie non-specific binding of the anti-VIII antibody)? Would you expect a GBS strain of another serotype to have similar ELISA readings compared to the cpsR or E mutant?

We do expect some background on the capsule ELISA due to non-specific binding of the polyclonal antibody as you suggest. The serotype VIII antibodies recognize polysaccharide and capsule is not the only polysaccharide on the surface of GBS. Particularly with a serotype VIII polyclonal antibody, it most likely relies on recognition of rhamnose to confer serotype specificity. This could possibly cause cross reactivity with the GBS group B carbohydrate which contains terminal rhamnose. When we have performed an ELISA with the serotype VIII antibody on a serotype V strain, there is a non-zero reading for that strain as well (see data below). As for the hydrophobicity assay, the paper we based our protocol on also showed wild-type GBS (serotype V, strain 10/84) to have over 50% of the bacteria bound to n-hexadecane(reference 76). This is a similar level of background to our data.

Reviewer #2 (Comments for the Author):

General comment

This is an interesting, important and well-executed study that highlights the potential for "vaccine escape mutants" to arise in the multi-host pathogen *Streptococcus agalactiae* or group B *Streptococcus* (GBS). The evolutionary evidence for the possibility of inter-species transfer and the functional evidence for the role of serotype VIII are convincing. The main flaw of the current write-up is that the authors cherry-pick from their results, apparently to align with their chosen focus on comparison with agricultural streptococci. Results, discussion and title need to be modified to accurately reflect results shown in Table S2, as detailed below.

Thank you for your thorough review and helpful critiques. We appreciate your focus on the overall story of the paper as you accurately pointed out that the streptococcal species from our BLASTp results are not strongly associated with an agricultural setting. This has led us to expand the discussion to include the relevance of GBS and other streptococci as multi-host pathogens with more opportunities to perform intra- and inter-species horizontal gene transfer. This has ultimately strengthened the paper and given it broader relevance. Below we address each point in more detail and explain how we have updated the manuscript to address the potential for genetic transfer between many streptococcal species beyond the two we tested in our functionality studies.

The abstract is clear and largely well-supported by data, with the exception of line 38. This line is incorrect in two ways, i.e., the species listed are not the closest matches (other species are equally as close based on Table S2 but not mentioned) and the niche of *S. suis* (mostly found in pigs) doesn't overlap with that of GBS (rarely found in pigs) unless the niche overlap occurs in humans infected with *S. suis*, which is a possibility (e.g., in China, where human *S. suis* occurs more commonly than anywhere else).

We have clarified line 43 by changing it to "proteins from streptococcal species that live in niches overlapping with GBS (including *S. suis* and *S. gallolyticus*) were closely related to CpsR." We hope this wording better reflects that there are other closely related species, and we are choosing to single out *S. suis* and *S. gallolyticus* to test *in vitro*. When referring to overlapping niches, we are indeed referencing humans as a potential point of contact for GBS and *S. suis* as you suggested. The role of *S. suis* as a zoonotic pathogen in Asia is explained in the discussion.

Importance section: Clear and informative, although comments about colonization, pathogenicity and vaccine candidates should be qualified as relating to human GBS. In bovine and piscine GBS, pseudogenisation of the capsular operon is observed (e.g., Almeida, Environ Microbiol. 2016 Nov;18(11):4216-4229. doi: 10.1111/1462-2920.13550) implying a lack of functional relevance, and GBS vaccines in fish and cattle are not based on CPS.

Thank you for pointing out this ambiguity. The first two sentences of the importance section were updated to specify that we are referring to GBS in humans. "Capsular

polysaccharide (CPS) is a key virulence factor that aids GBS in colonization and pathogenicity in humans. The major human vaccine candidate against GBS is a CPS-based vaccine including six serotypes.”

The introduction is very clear and very well-referenced.

Genomic and laboratory-based methods are informative and elegant.

The results section is very accessible and nicely illustrated but appears to lack objectivity. In Table 2, authors cherry-picked from the results presented in Table S2. For *cpsE*, scores for *S. orisratti* and *S. suis* were identical yet *S. orisratti* is not mentioned, maybe because it isn't an "agricultural streptococcus"? Likewise, for *cpsJ*, scores are identical for *S. equinus* and *S. gallolyticus* yet only *S. equinus* is listed in Table 1. Indeed, authors state (line 205-206) that donor and recipient have had time to evolve since a historical genetic transfer. Why, then, do authors not consider other potential donors that are close matches, such as *S. thermophilus* (CpsR), *S. lutetiensis* (CpsH) or *S. infantarius* (CpsJ)? The entire manuscript, including Table 2, results as presented in line 133-140, Figure 2A, and all related discussion as well as the manuscript title and abstract need to be updated to reflect results from Table S2, which do not support the association with agricultural streptococci as *Streptococcus orisratti* is not an agriculture-related species and neither is *Strep. gallolyticus*. While the latter has certainly been reported from agricultural animals, e.g. turkeys and dairy cattle, it is more commonly associated with humans, particularly with colorectal cancer (e.g. Gut Microbiome (Camb). 2024 Nov 4;5:e9. doi: 10.1017/gmb.2024.11.) and invasive disease in infants (e.g., *Pediatr Infect Dis J.* 2022 Nov 1;41(11):e494-e497. doi: 10.1097/INF.0000000000003682), putting it in the same niches as GBS in humans. While the selective presentation of results does not affect the core premise of the manuscript, i.e. that interspecies transfer may contribute to evolution of GBS capsular serotypes with functional significance, it does affect the surrounding narrative, e.g. the title and abstract (as already stated) and the conclusion drawn in line 141-142.

Thank you for identifying the absence of top BLASTp hits with identical percent identities to the species listed in Table 2. *S. orisratti* and *S. gallolyticus* have been added to Table 2 to correct this.

We also appreciate the thorough analysis of the totality of the BLASTp results, and agree that all of these species would have been interesting to investigate. Ultimately, we focused on *S. suis* and *S. gallolyticus* because not only were they top hits for several genes in the variable part of the *cps* locus of serotype VIII, but they were also present in the top 100 hits for each protein, CpsERHJ. *S. orisratti* was not found in the top 100 matches for CpsH or CpsJ. *S. thermophilus* was only found in CpsR BLAST, and *S. infantarius* was not found in the CpsH BLAST. *S. suis* is in all BLAST results, and was most similar to CpsR, the main protein we were interested in, so this made it the best candidate for follow up. To be fair, it is a more studied species of *Streptococcus* so it is possible that the database is biased. Lastly, *S. lutetiensis* is in all BLAST results

besides CpsK as is *S. gallolyticus*, but *S. gallolyticus* was the top hit on 2 out of the 4 genes of interest making it our second-choice candidate to test. We picked two species for convenience of testing *in vitro*, but certainly many of these other species with CpsR orthologs could have been tested, and it is possible those genes could complement the function of *cpsR* in GBS as well. We focused on *S. suis* and *S. gallolyticus* as examples of the potential for genetic transfer from other streptococcal species at large.

The decision to title this paper with the term “agricultural streptococci” was made after picking these candidates to test, but we agree that focusing solely on these two streptococcal species when titling the paper was inconsistent with the full BLAST results and have therefore changed the title to: “Expansion of the Group B *Streptococcus* serotype repertoire via gene acquisition from other streptococcal species”

We also changed lines 149-152 to better reflect how we made the decision to pick these two species over other candidates “Based on these results, we focused on *S. suis* and *S. gallolyticus* as potential candidates for a genetic transfer event. *S. suis* and *S. gallolyticus* were selected because they had the most similar sequences for two out of the four Cps proteins and were present in the top 100 alignments for all four proteins.”

From Figure 2C, it is difficult to ascertain whether the GBS sequence is more closely related to *S. suis* or *S. ruminantium*. The branch lengths of the tree rather than the positioning of the coloured leaves of the tree are the salient information here and insufficient detail about branch lengths is provided to assess which one is closer. The close proximity of *S. gallolyticus* to GBS CpsR, as described in line 162-164, is not supported by Figure 2C.

We appreciate your comment identifying the lack of information about branch lengths in the tree. We accidentally omitted the scale bars, and these have been restored in Figure 2B and C. Also, a more complete description of how the tree was constructed has been added to the methods section “The evolutionary history was inferred by using the Maximum Likelihood method and JTT matrix-based model(71). The tree with the highest log likelihood (-42184.15) is shown. Initial tree(s) for the heuristic search were obtained automatically by applying Neighbor-Join and BioNJ algorithms to a matrix of pairwise distances estimated using the JTT model, and then selecting the topology with superior log likelihood value. The tree is drawn to scale, with branch lengths measured in the number of substitutions per site. This analysis involved 821 amino acid sequences. There were a total of 556 positions in the final dataset. Evolutionary analyses were conducted in MEGA11(72,73).” In addition, further details about the branch lengths have been added to the figure legend “The tree is drawn to scale, and branch lengths correspond to the number of substitutions per site.”

To make our wording more accurate when describing the Fig. 2C tree we have changed the wording in line 185-187 from “Additional sequences from *S. suis* and *S. gallolyticus* are both in close proximity to GBS CpsR, along with sequences from other streptococcal species.” to “Additional sequences from *S. suis* and *S. gallolyticus* both

appear in the subsetted tree in Fig. 2C GBS CpsR, along with sequences from other streptococcal species.”

In Figure 3, panels C and D appear to have been inverted. Text for figure 3C (line 193) refers to the image in panel D, and text for figure 3D (line 198) refers to the image currently presented in panel C.

Thank you for pointing this out, the notation in the text has been fixed.

In lines 198 and 265, authors state that the cpsR mutant phenocopied the acapsular strain. Figure 3, however, appears to show non-overlapping error bars for those two mutants and a P-value is, intentionally or unintentionally, withheld, so while resemblance is demonstrated, with the data provided, phenocopying is not.

While there are no overlapping error bars, there was no statistical significance found between the hydrophobicity of Δ cpsR and Δ cpsE. The p value is 0.1340.

Line 208 asserts that authors wish to understand the potential for genetic transfer. This is not assessed in the current study. Rather, authors assess whether the phenotype of one species can be induced with genetic material from another species in an in vitro system. Whilst I don't dispute the evidence they provide in support of this in vitro phenomenon, nor the hypothesis relating to HGT (line 217-218), genetic transfer itself is not tested. Differences in HGT may exist between and within gram-positive bacterial species and the potential for HGT between GBS and the suggested species, or other species that may be equally as relevant based on Table 2S, is not assessed. Likewise, creation of mutants and revertants and demonstration of their functional impacts is no evidence of an evolutionary relationship (line 266).

These are good points, and we have updated our language to be more accurate to what we are specifically testing. Lines 233-235 have been changed from “To gain understanding of the potential for genetic transfer” to, “To understand if genetic material from another streptococcal species is sufficient to complement the production of serotype VIII capsule in a cpsR deficient strain of GBS.”

Also, lines 302-303 “To investigate a potential evolutionary relationship” has been changed to, “To investigate the functionality of potentially transferred genetic material.”

The discussion opens with three mechanisms for serotype replacement (lines 236-238). Could a reference be provided for each of those mechanisms? Please also add a numbered reference for Roy et al. (line 276). Authors cite Roy et al. as stating “Another striking feature of these two pathogens is that *S. suis* and GBS are the only Gram-positive bacteria expressing a sialylated CPS.” This is indeed stated in Roy et al., but it is not supported by references in their paper and it is not clear whether all other potential donors (e.g. *S. equinus*, *gallolyticus*, *infantarius*, *orisratti* or *thermophilus*, based on Table S2) have been considered. Would the authors be able to comment on whether there is absence of evidence for those species, or evidence of absence?

The discussion opens with a theoretical statement about the potential for serotype replacement and the introductory line has been updated from “Serotype replacement can occur via three mechanisms” to “Serotype replacement has the potential to occur via three mechanisms.” The evidence for mechanism one and potential for mechanisms two and three is cited in detail throughout the rest of the paragraph.

Thank you for pointing out the missing numbered reference, this has been added to the text.

The presence of sialylated capsule on *S. suis* and GBS is an interesting relationship, but we would like to clarify that this section of the discussion was not intended to show more evidence that there was a genetic transfer event from *S. suis* to GBS. Rather, it was to show that GBS can use genetic material from other streptococcal species. GBS and *S. suis* are the only two confirmed streptococcal species with sialic acid in their capsule, but the reviewer is correct to point out that this is an absence of evidence because surely not every serotype of every encapsulated streptococcal species has been identified. Importantly for this paper, if other untested species also produced sialylated capsule this would not make them better or worse candidates for testing a potential genetic relationship to GBS because the sialic acid synthesis section of the GBS *cps* locus (*neuBCDA*) is well conserved even within serotype VIII. Therefore, we do not think the sialic acid section of the locus was transferred to make this serotype. In addition, when BLAST was performed on the GBS sialyltransferase CpsK, all percent identities were below 42% and the only streptococcal species hit were *S. suis*, *Streptococcus* sp. FSL R7-0212, and *S. uberis* (which has a hyaluronic acid capsule).

In I. 286, authors appear to conflate potential (ability to do something, which would be a trait of the bacteria) with opportunity (circumstances allowing for the activity, such as a shared niche). Opportunity for genetic transfer seems most likely in the human gut, where GBS and SBSEC co-exist.

This is a good point, “potential” has been swapped for “opportunity” in line 325.

L. 294-296 GBS also cause disease in fishes, camels, frogs and many other species, including farmed and wild species. Why selectively cite its occurrence in dairy cattle?

The discussion lacks consideration of potentially relevant streptococcal species other than *S. suis* and *S. gallolyticus* as listed in table S2. Several of those species, certainly members of the SBSEC, are multi-host pathogens rather than “animal streptococci” as suggested in line 328. Please amend the discussion accordingly.

We appreciate this point and agree that the host range of GBS is important to consider. We have expanded the discussion to include how this wider host range allows GBS more locations and opportunities to contact other streptococcal species and swap DNA. This expanded section also includes potential overlap with the other species from our

BLAST results and considers the implications of GBS and other streptococcal species being multi-host pathogens. This is the new paragraph (lines 337-349):

“GBS is a multi-host bacterial pathogen infecting fish, frogs, and camels in addition to humans and cows(59). Given its broader host range, GBS has the opportunity to overlap with many streptococcal species in addition to the two we tested. This includes many other species from our BLASTp results, particularly other multi-host bacterial pathogens such as the other SBSEC species or *Streptococcus ruminantium* which is in close proximity to GBS on our CpsR phylogenetic tree but did not show up in our BLASTp results most likely because it was recently reclassified from *S. suis*(60). Additionally, lineages of GBS that can infect multiple hosts have higher genome plasticity when compared to host specialist lineages of GBS(59). This suggests that GBS strains capable of colonizing multiple hosts have an increased likelihood of HGT and may have potential to create new serotypes.”

Re: Spectrum01227-25R1 (Expansion of the Group B *Streptococcus* serotype repertoire via gene acquisition from other streptococcal species)

Dear Dr. Adam J. Ratner:

I am glad to accept your paper for publication in Microbiology Spectrum. Please take into account two minor points raised by the reviewer 2 below.

Your manuscript has been accepted, and I am forwarding it to the ASM production staff for publication. Your paper will first be checked to make sure all elements meet the technical requirements. ASM staff will contact you if anything needs to be revised before copyediting and production can begin. Otherwise, you will be notified when your proofs are ready to be viewed.

Sincerely,
Nicolas Soler
Editor
Microbiology Spectrum

Reviewer #1 (Comments for the Author):

The revised manuscript adequately addresses my prior minor critiques.

Reviewer #2 (Comments for the Author):

The authors have been as meticulous and rigorous in their response to reviewer comments as they were in their original evolutionary and genetic studies, and almost all concerns about data interpretation have been addressed. The updated and expanded discussion helps to explain the importance of the authors' findings. Minor points for clarification are listed below:

Line 77, Line 81, Line 85 - "colonization rates" of 20% or 35.6% are suggested for non-vaccine serotypes and serotype VIII, respectively. Presumably, authors mean to say that of the colonized patients, 20% or 35.6%, respectively, are colonized with non-vaccine serotypes and serotype VIII. Considering that GBS colonisation rates in humans are, on average, around 20%, which would comprise all serotypes, colonization of 20% of the population with non-vaccine serotype alone (or of an even larger population with serotype VIII) would seem highly unlikely.

Line 297-302 - The niche overlap between *S. gallolyticus* and GBS in animals is not clearly explained. While there is obvious niche overlap between the two streptococcal species in the human gut, GBS is almost never found in horses so niche overlap in this host species seems unlikely. For cattle, are the authors suggesting that milk consumption, like consumption of pork with *S.*

suis, could lead to introduction of bacteria, in this case animal GBS, into the human GI tract where it would have a niche that overlaps with resident *S. gallolyticus*? Or are authors suggesting niche overlap in the bovine GI tract? Although GBS is primarily a mastitis pathogen in cattle, in line with the authors' comment in line 301/302, it has also been detected in bovine faeces (doi: 10.1016/j.vetmic.2015.12.014; doi: 10.1371/journal.pone.0008795), so this may be a possibility.